# Diffusion Maps for Textual Network Embedding

**Xinyuan Zhang, Yitong Li, Dinghan Shen, Lawrence Carin**
Department of Electrical and Computer Engineering
Duke University
Durham, NC 27707
{xy.zhang, yitong.li, dinghan.shen, lcarin}@duke.edu

## Abstract

Textual network embedding leverages rich text information associated with the network to learn low-dimensional vectorial representations of vertices. Rather than using typical natural language processing (NLP) approaches, recent research exploits the relationship of texts on the same edge to graphically embed text. However, these models neglect to measure the *complete* level of connectivity between any two texts in the graph. We present diffusion maps for textual network embedding (DMTE), integrating global structural information of the graph to capture the semantic relatedness between texts, with a diffusion-convolution operation applied on the text inputs. In addition, a new objective function is designed to efficiently preserve the high-order proximity using the graph diffusion. Experimental results show that the proposed approach outperforms state-of-the-art methods on the vertex-classification and link-prediction tasks.

## 1 Introduction

Learning effective vectorial embeddings to represent text can lead to improvements in many natural language processing (NLP) tasks. However, most text embedding models do not embed the semantic relatedness between different texts. Graphical text networks address this problem by adding edges between correlated text vertices. For example, paper citation networks contain rich textual information and the citation relationships provide structural information to reflect the similarity between papers. Graphical text embedding naturally extends the problem to network embedding (NE), mapping vertices of a graph into a low-dimensional space. The

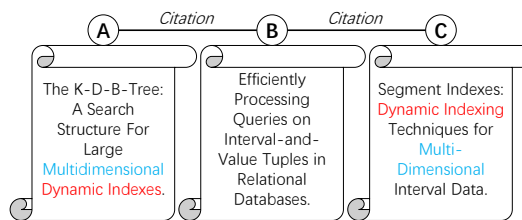

Figure 1: Three sentences from the *DBLP* dataset. Vertices A and C are second neighbors, *i.e.*, two vertices that are not on the same edge but share at lease one common neighbor (vertex B). The alignment words are colored.

learned representations containing structure and textual information can be used as features for network tasks, such as vertex classification [22], link prediction [14], and tag recommendation [31]. Learning network embeddings is a challenging research problem, due to the sparsity, non-linearity and high dimensionality of the graph data.

In order to exploit textual information associated with each vertex, some NE models [13, 33, 19, 26] embed texts with a variety of NLP approaches, ranging from bag-of-words models to deep neural models. However these text embedding methods fail to consider the semantic distance indicated from the graph. In [30, 24] it was recently proposed to simultaneously embed two texts on the same edge using a mutual-attention mechanism. But in real-world sparse networks, it is intuitive that two connected vertices do not necessarily share more similarities than two unconnected vertices. Figure 1

presents three examples from the *DBLP* dataset. By aligning *dynamic index* and *multi-dimensional*, the sentences of vertex A and vertex C are closer than the sentence of their common first neighbor, vertex B. The relatedness between two vertices that are not linked by an edge cannot be preserved by only capturing the local pairwise proximity.

We propose a flexible approach for textual network embedding, including global structural information without increasing model complexity. Global structure information serves to capture the long-distance relationship between two texts, incorporating connection paths within different steps. The diffusion-convolution operation [2] is employed to build a latent representation of the graph-structured text inputs, by scanning a diffusion map across each vertex. The graph diffusion, comprised of a normalized adjacency matrix and its power series, provides the probability of random walks from one vertex to another within a certain number of steps in the graph. The idea is to measure the level of connectivity between any two texts when considering all paths between them. In this study, we consider text-based information networks, but our model can be flexibly extended to other types of content.

We further use the graph diffusion to redesign the objective function, capturing high-order proximity. Unlike some NE models [27], that calculate the probability of vertex $v_i$ being generated by $v_j$, we preserve high-order proximity by calculating the probability of vertex $v_i$ given the diffusion map of $v_j$. Compared to GraRep [5], the proposed objective function is more computationally efficient, especially for large-scale networks, because it does not need matrix factorization during training. This objective function is able to scale to directed or undirected, and weighted or unweighted graphs.

To demonstrate the effectiveness of our model, we focus on two common tasks in analysis of textual information networks: $(i)$ multi-label classification, where we predict the labels of each text; and $(ii)$ link prediction, where we predict the existence of an edge given a pair of vertices. The experiments are conducted on several real-world datasets of information networks. Experimental results show that the DMTE model outperforms all other methods considered. The superiority of the proposed approach indicates that the diffusion process helps to incorporate long-distance relationship between texts and thus to achieve more informative textual network embeddings.

## 2   Related Work

**Text Embedding**    Many existing methods embed text messages into a vector space for various NLP tasks. Early approaches include bag-of-words models or topic models [4]. The Skip-gram model [16], which learns distributed word vectors by utilizing word co-occurrences in a local context, has been further extended to the document level via a paragraph vector [13] to learn text latent representations. To exploit the internal structure of text, more-complicated text embedding models have emerged, adopting deep neural network architectures. For example, convolutional neural networks (CNNs) [10, 6, 34] have been considered to apply a convolution kernel over different positions of the text, followed by max-pooling to obtain a fixed-length vectorial representation. Recursive neural tensor networks (RNTNs) [25] have applied a tensor-based composition function over parse trees to obtain sentence representations. LSTM-based recurrent neural networks (RNNs) [12] capture long-term dependencies in the text, using long short-term memory cells. However, deep neural architectures usually assume the availability of a large dataset, unrealistic for many information networks. When the data size is small, some methods [18, 9] avoid over-fitting by simply averaging embeddings of each word in the text, achieving competitive empirical results.

**Network Embedding**    Earlier works including IsoMap [29], LLE [21], and Laplacian Eigenmaps [3] transform feature vectors of vertices into an affinity graph, and then solve for the leading eigenvectors as the embedding. Recent NE models focus on learning the vectorial representation of existing networks. For example, DeepWalk [20] uses the Skip-gram model [16] on vertex sequences generated by truncated random walks, learning vertex embeddings. In node2vec [8], the random walk strategy of DeepWalk is modified for multi-scale representation learning. To exploit the distance between vertices, LINE [27] designed objective functions to preserve the first-order and second-order proximity, while [5] integrates global structure information by expanding the proximity into $k$-order. In [32] deep models are employed to capture the nonlinear network structure. However, all these methods only consider structural information of the network, without leveraging rich heterogeneous information associated with vertices; this may result in less informative representations, especially when the edges are sparse.

To address this issue, some recent works combine structure and content information to learn better embeddings. For example, TADW [33] shows that DeepWalk is equivalent to matrix factorization, and text features can be incorporated into the framework. TriDNR [19] uses information from structure, content and labels in a coupled neural network architecture, to learn the vertex representation. CENE [26] integrates text modeling and structure modeling by regarding the content information as a special kind of vertex. CANE [30] learns two embedding vectors for each vertex where the context-aware text embedding is obtained using a mutual attention mechanism. However, none of these methods takes into account the similarities of context influenced by global structural information.

## 3 Problem Definition

**Definition 1.** A ***textual information network*** is $G = (V, E, T)$, where $V = \{v_i\}_{i=1,\cdots,N}$ is the set of vertices, $E = \{e_{i,j}\}_{i,j=1}^{N}$ is the set of edges, and $T = \{t_i\}_{i=1,\cdots,N}$ is the set of texts associated with vertices. Each edge $e_{i,j}$ has a weight $s_{i,j}$ representing the relationship between vertices $v_i$ and $v_j$. If $v_i$ and $v_j$ are not linked, $s_{i,j} = 0$. If there exists an edge between $v_i$ and $v_j$, $s_{i,j} = 1$ for an unweighted graph, and $s_{i,j} > 0$ for a weighted graph. A *path* is a sequence of edges that connect two vertices. The text of vertex $v_i$, $t_i$, is comprised of a word sequence $< w_1, \cdots, w_{|t_i|} >$.

**Definition 2.** Let $\mathbf{S} \in \mathbb{R}^{N \times N}$ be the adjacency matrix of a graph whose entry $s_{i,j} \geq 0$ is the weight of edge $e_{i,j}$. The ***transition matrix*** $\mathbf{P} \in \mathbb{R}^{N \times N}$ is obtained by normalizing rows of $\mathbf{S}$ to sum to one, with $p_{i,j}$ representing the transition probability from vertex $v_i$ to vertex $v_j$ within one step. Then an $h$-step transition matrix can be computed with $\mathbf{P}$ to the $h$-th power, *i.e.*, $\mathbf{P}^h$. The entry $p_{i,j}^h$ refers to the transition probability from vertex $v_i$ to vertex $v_j$ within exactly $h$ steps.

**Definition 3.** A ***network embedding*** aims to learn a low-dimensional vector $\boldsymbol{v}_i \in \mathbb{R}^d$ for vertex $v_i \in V$, where $d \ll |V|$ is the dimension of the embedding. The embedding matrix $\mathbf{V}$ for the complete graph is the concatenation of $\{\boldsymbol{v}_1, \boldsymbol{v}_2, \cdots, \boldsymbol{v}_N\}$. The distance between vertices on the graph and context similarity should be preserved in the representation space.

**Definition 4.** The ***diffusion map*** of vertex $v_i$ is $\boldsymbol{u}_i$, the $i$-th row of the diffusion embedding matrix $\mathbf{U}$, which maps from vertices and their embeddings to the results of a diffusion process that begins at vertex $v_i$. $\mathbf{U}$ is computed by

$$\mathbf{U} = \sum_{h=0}^{H-1} \lambda_h \mathbf{P}^h \mathbf{V}, \tag{1}$$

where $\lambda_h$ is the importance coefficient that typically decreases as the value of $h$ increases. The high-order proximity in the network is preserved in diffusion maps.

## 4 Method

We employ a diffusion process to build long-distance semantic relatedness in text embeddings, and global structural information in the objective function. To incorporate both the structure and textual information of the network, we adopt two types of embeddings $\boldsymbol{v}_i^s$ and $\boldsymbol{v}_i^t$ for each $v_i$ vertex, as proposed in [30]. The structure-based embedding vector $\boldsymbol{v}_i^s$ is obtained by feeding the $i$-th row of a learned structure embedding table $\mathbf{E}_s \in \mathbb{R}^{N \times d_s}$ into a function. The text-based embedding vector $\boldsymbol{v}_i^t$ is obtained by applying the diffusion convolutional operation on the text inputs (see Section 4.2). Here dimensions of the structure embedding and the text embedding satisfy $d_s + d_t = d$. The embedding of vertex $v_i$ is simply the concatenation of $\boldsymbol{v}_i^t$ and $\boldsymbol{v}_i^s$, *i.e.*, $\boldsymbol{v}_i = \boldsymbol{v}_i^t \oplus \boldsymbol{v}_i^s$. In this work, $\boldsymbol{v}_i$ is learned by an unsupervised approach, and it can be used directly as a feature vector of vertex $v_i$ for various tasks. The objective function consists of four parts, which measure both the structure and text embeddings. The high-order proximity is preserved during training without increasing computational complexity. The entire framework for textual network embedding is illustrated in Figure 3 where each vertex is associated with a text.

### 4.1 Diffusion Process

Initially the network only has a few active vertices, due to sparsity. Through the diffusion process, information is delivered from active vertices to inactive ones by filling information gaps between vertices [1]; vertices may be connected by indirect, multi-step paths. This process is the same as the molecular diffusion in a fluid, where particles move from high-concentration areas to low-concentration areas. We introduce the transition matrix $\mathbf{P}$ and its power series for the diffusion process. The directed graph with four vertices and normalized weights in Figure 2 shows the smoothing effect of the high order of $\mathbf{P}$ in diffusion process. The original graph only has edges $e_{1,2}$, $e_{1,3}$, $e_{3,4}$ and $e_{1,4}$, while the in-

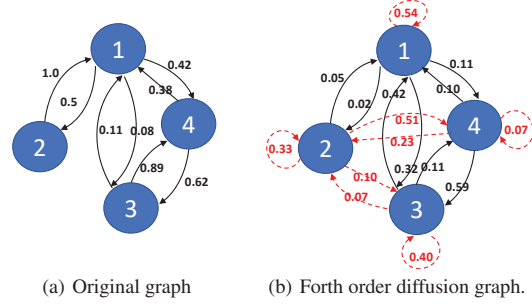

(a) Original graph      (b) Forth order diffusion graph.

Figure 2: A simple example of diffusion process in a directed graph.

formation gaps between other vertices are not depicted. The diffusion process can smooth the whole graph with the higher order of $\mathbf{P}$, so that indirect relationships, such as $(n_2, n_4)$, can be connected (via a multi-step diffusion process). As we can see from Figure 2(b), the fourth-order diffusion graph is fully connected. The number associated with each edge represents the transition probability from one vertex to another within exactly 4 steps. The network will be stable when information is eventually evenly distributed.

## 4.2 Text Embedding

A word sequence $t = <w_1, \cdots, w_{|t|}>$ is mapped into a set of $d_t$-dimensional real-valued vectors $<\boldsymbol{w}_1, \cdots, \boldsymbol{w}_{|t|}>$ by looking up the word embedding matrix $\mathbf{E}_w$. Here $\mathbf{E}_w \in \mathbb{R}^{|w| \times d_t}$ is initialized randomly, and learned during training, and $|w|$ is the vocabulary size of the dataset. We can obtain a simple text representation $\boldsymbol{x}_i \in \mathbb{R}^{d_t}$ of vertex $v_i$ by taking the average of word vectors. Although the word order is not preserved in such a representation, taking the average of word embeddings can avoid over-fitting efficiently, especially when the data size is small [23]. Given the fixed-length vectors of each text, the input texts can be represented by matrix $\mathbf{X} \in \mathbb{R}^{N \times d_t}$, where the $i$-th row is $\boldsymbol{x}_i$.

$$\boldsymbol{x} = \frac{1}{|t|} \sum_{i=1}^{|t|} \boldsymbol{w}_i, \qquad \mathbf{X} = \boldsymbol{x}_1 \oplus \boldsymbol{x}_2 \oplus \cdots \oplus \boldsymbol{x}_N. \qquad (2)$$

Alternatively, we can use the bi-directional LSTM [7] which processes a text from both directions to capture long-term dependencies. Text inputs are represented by the mean of all hidden states.

$$\overrightarrow{\boldsymbol{h}}_i = LSTM(w_i, \boldsymbol{h}_{i-1}), \qquad \overleftarrow{\boldsymbol{h}}_i = LSTM(w_i, \boldsymbol{h}_{i+1}) \qquad (3)$$

$$\boldsymbol{x} = \frac{1}{|t|} \sum_{i=1}^{|t|} (\overrightarrow{\boldsymbol{h}}_i \oplus \overleftarrow{\boldsymbol{h}}_i), \qquad \mathbf{X} = \boldsymbol{x}_1 \oplus \boldsymbol{x}_2 \oplus \cdots \oplus \boldsymbol{x}_N. \qquad (4)$$

However, in this text representation matrix for both approaches, the embeddings are completely independent, without leveraging the semantic relatedness indicated from the graph. To address this issue, we employ the diffusion convolutional operator [2] to measure the level of connectivity between any of two texts in the network.

Let $\mathbf{P}^* \in \mathbb{R}^{N \times H \times N}$ be a tensor containing $H$ hops of power series of $\mathbf{P}$, *i.e.*, the concatenation of $\{\mathbf{P}^0, \mathbf{P}^1, \cdots, \mathbf{P}^{H-1}\}$. $\mathbf{V}_t^* \in \mathbb{R}^{N \times H \times d}$ is the tensor version of the text embedding representation, after the diffusion convolutional operation. The activation $\mathbf{V}_t^{*(i,j,k)}$ for vertex $i$, hop $j$, and feature $k$ is given by

$$\mathbf{V}_t^{*(i,j,k)} = f\left(\mathbf{W}^{(j,k)} \cdot \sum_{n=1}^{N} \mathbf{P}^{*(i,j,n)} \mathbf{X}^{(n,k)}\right), \qquad (5)$$

where $\mathbf{W} \in \mathbb{R}^{H \times d}$ is the weight matrix and $f$ is a nonlinear differentiable function. The activations can be expressed equivalently using tensor notation

$$\mathbf{V}_t^* = f(\mathbf{W} \odot \mathbf{P}^* \mathbf{X}), \qquad (6)$$

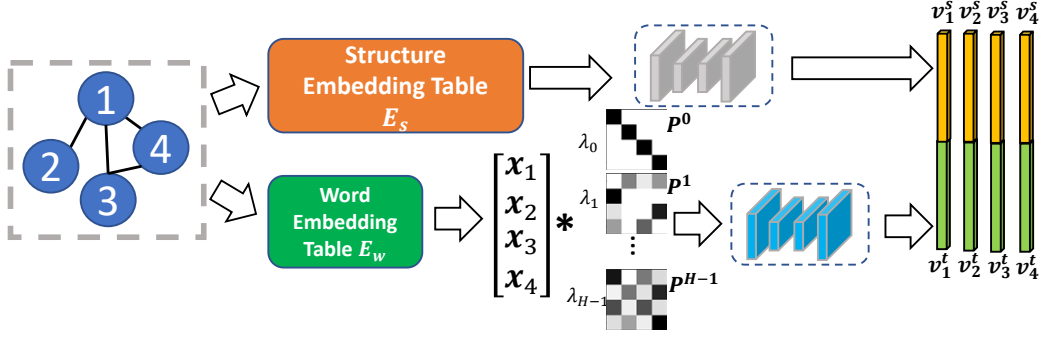

Figure 3: An illustration of our framework for textual network embedding.

where $\odot$ represents element-wise multiplication. This tensor representation considers all paths between two texts in the network, and thus includes long-distance semantic relationship. With longer paths discounted more than shorter paths, the text embedding matrix $\mathbf{V}_t$ is given by

$$\mathbf{V}_t = \sum_{h=0}^{H-1} \lambda_h \mathbf{V}_t^{*(:,h,:)}. \tag{7}$$

Through the diffusion process, text representations, *i.e.*, rows of $\mathbf{V}_t$ are not embedded independently. With the whole graph being smoothed, indirect relationships between texts that are not on the same edge can be considered to learn embeddings.

### 4.3 Objective Function

Given the set of edges $E$, the goal of DMTE is to maximize the following overall objective function:

$$\mathcal{L} = \sum_{e \in E} L(e) = \sum_{e \in E} \alpha_{tt} L_{tt}(e) + \alpha_{ss} L_{ss}(e) + \alpha_{st} L_{st}(e) + \alpha_{ts} L_{ts}(e) \tag{8}$$

where $\alpha_{tt}$, $\alpha_{ss}$, $\alpha_{st}$, and $\alpha_{ts}$ control the weight of corresponding objectives. The overall objective consists of four parts: $L_{tt}(e)$ denotes the objective for text embeddings, $L_{ss}(e)$ denotes the objective for structure embeddings, $L_{st}(e)$ and $L_{ts}(e)$ denote the objectives that consider both structure and text embeddings to map them into the same representation space. We assume the network is directed, since the undirected edge can be considered as two opposite-directed edges with equal weights. Then each objective is to measure the log-likelihood of generating $v_i$ conditioned on $v_j$, where $v_i$ and $v_j$ are on the same directed edge:

$$L_{tt}(e) = s_{i,j} \log p(\boldsymbol{v}_i^t | \boldsymbol{v}_j^t) = s_{i,j} \log \frac{\exp(\boldsymbol{v}_i^t \cdot \boldsymbol{v}_j^t)}{\sum_{\boldsymbol{v}_k^t \in \mathbf{V}_t} \exp(\boldsymbol{v}_k^t \cdot \boldsymbol{v}_j^t)}, \tag{9}$$

$$L_{ss}(e) = s_{i,j} \log p(\boldsymbol{v}_i^s | \boldsymbol{u}_j^s) = s_{i,j} \log \frac{\exp(\boldsymbol{v}_i^s \cdot \boldsymbol{u}_j^s)}{\sum_{\boldsymbol{v}_k^s \in \mathbf{V}_s} \exp(\boldsymbol{v}_k^s \cdot \boldsymbol{u}_j^s)}, \tag{10}$$

$$L_{st}(e) = s_{i,j} \log p(\boldsymbol{v}_i^s | \boldsymbol{v}_j^t) = s_{i,j} \log \frac{\exp(\boldsymbol{v}_i^s \cdot \boldsymbol{v}_j^t)}{\sum_{\boldsymbol{v}_k^s \in \mathbf{V}_s} \exp(\boldsymbol{v}_k^s \cdot \boldsymbol{v}_j^t)}, \tag{11}$$

$$L_{ts}(e) = s_{i,j} \log p(\boldsymbol{v}_i^t | \boldsymbol{u}_j^s) = s_{i,j} \log \frac{\exp(\boldsymbol{v}_i^t \cdot \boldsymbol{u}_j^s)}{\sum_{\boldsymbol{v}_k^t \in \mathbf{V}_t} \exp(\boldsymbol{v}_k^t \cdot \boldsymbol{u}_j^s)}. \tag{12}$$

Note that $p(\cdot | \boldsymbol{u}_j^s)$ computes the probability conditioned on the diffusion map of vertex $v_j$, and $p(\cdot | \boldsymbol{v}_j^t)$ computes the probability conditioned on the text embedding of vertex $v_j$. Compared to using $\boldsymbol{v}_j^s$ to compute the conditional probability, the diffusion map $\boldsymbol{u}_j^s$ utilizes both local information and global relations of vertex $v_j$ in the graph. We use $\boldsymbol{v}_j^t$ instead of the diffusion map $\boldsymbol{u}_j^t$ because the global structural information is included during text embedding, with the diffusion convolutional operation. Moreover the high-order proximity is preserved without using matrix factorization, which may be computationally inefficient for large-scale networks.

### 4.4 Optimization

Optimizing (8) is computationally expensive, since the conditional probability requires the summation over the entire vertex set. In [17] negative sampling was proposed to solve this problem. For each edge $e_{i,j}$, we sample multiple negative edges according to some noisy distribution. Then during training the conditional function $p(\boldsymbol{v}_i|\boldsymbol{v}_j)$ can be replaced by

$$\log \sigma(\boldsymbol{v}_i \cdot \boldsymbol{v}_j) + \sum_{k=1}^{K} E_{\boldsymbol{v}_k \sim P_n(v)}[\log \sigma(-\boldsymbol{v}_k \cdot \boldsymbol{v}_j)], \tag{13}$$

where $\sigma(\cdot)$ is the sigmoid function, $K$ is the number of negative samples, and $P_n(v) \propto d_v^{3/4}$ is the distribution of vertices with $d_v$ being the out-degree of vertex $v$. All parameters are jointly trained. Adam [11] is adopted for stochastic optimization. In each step, Adam samples a mini-batch of edges and then updates the model parameters.

## 5 Experiments

We evaluate the proposed method for the multi-label classification and link prediction tasks. We design four versions of DMTE in our experiments: ($i$) DMTE without diffusion process; ($ii$) DMTE with text embedding only; ($iii$) DMTE with bidirectional LSTM (Bi-LSTM); ($iv$) DMTE with word average embedding (WAvg). In DMTE without diffusion process, the diffusion convolutional operation is not added on top of the text inputs, *i.e.*, the text embedding matrix $\mathbf{V}_t$ is directly replaced by $\mathbf{X}$ in Eq. 2. In DMTE with text embedding only, the embedding of vertex $v_i$ is only $\boldsymbol{v}_i^t$ instead of the concatenation of $\boldsymbol{v}_i^t$ and $\boldsymbol{v}_i^s$. In DMTE with Bi-LSTM, the input texts embedding matrix $\mathbf{X}_t$ is obtained using Eq. 4. In DMTE with WAvg, the input texts embedding matrix $\mathbf{X}_t$ is obtained using Eq. 2. We compare the four versions of DMTE model with seven competitive network embedding algorithms. Experimental results for multi-label classification are evaluated by Macro F1 scores and experimental results for link prediction are evaluated by Area Under the Curve (AUC).

**Datasets**  We conduct experiments on three real-world datasets: DBLP, Cora, and Zhihu.

- DBLP [28] is a citation network that consists of bibliography data in computer science. In our experiments, 60744 papers are collected in 4 research areas: database, data mining, artificial intelligence, and computer vision. The network has 52890 edges indicating the citation relationship between papers.

- Cora [15] is a citation network that consists of 2277 machine learning papers in 7 classes. The network has 5214 edges indicating the citation relationship between papers.

- Zhihu [26] is a Q&A based community social network in China. In our experiments, 10000 active users are collected as vertices and 43894 edges indicating the relationship. The description of their interested topics are used as text information.

**Baselines**  The following baselines are compared with our DMTE model:

- Structure-Based Methods: DeepWalk [20], LINE [27], node2vec [8].
- Structure and Text Combined Methods: TADW [33], Tri-DNR [19], CENE [26], CANE [30].

**Evaluation and Parameter Settings**  For link prediction, we evaluate the performance with AUC, which is widely used for a ranking list. Since the testing set only contains existing edges as positive instances, we randomly sample the same number of non-existing edges as negative instances. Positive and negative edges are ranked according to a prediction function and AUC is employed to measure the probability that vertices on a positive edge are more similar than those on a negative edge. The experiment for each training ratio is executed 10 times and the mean AUC scores are reported, where the higher value indicates a better performance.

For multi-label classification, we evaluate the performance with Macro-F1 scores. We first learn embeddings with all edges and vertices in an unsupervised way. Once the vertex embeddings are obtained, we feed them into a classifier. The experiment for each training ratio is executed 10 times and the mean Macro-F1 scores are reported where the higher value indicates a better performance.

Table 1: AUC scores for link prediction on Cora.

| % of edges | 15% | 25% | 35% | 45% | 55% | 65% | 75% | 85% | 95% |
|---|---|---|---|---|---|---|---|---|---|
| Deep Walk | 56.0 | 63.0 | 70.2 | 75.5 | 80.1 | 85.2 | 85.3 | 87.8 | 90.3 |
| LINE | 55.0 | 58.6 | 66.4 | 73.0 | 77.6 | 82.8 | 85.6 | 88.4 | 89.3 |
| node2vec | 55.9 | 62.4 | 66.1 | 75.0 | 78.7 | 81.6 | 85.9 | 87.3 | 88.2 |
| TADW | 86.6 | 88.2 | 90.2 | 90.8 | 90.0 | 93.0 | 91.0 | 93.4 | 92.7 |
| TriDNR | 85.9 | 88.6 | 90.5 | 91.2 | 91.3 | 92.4 | 93.0 | 93.6 | 93.7 |
| CENE | 72.1 | 86.5 | 84.6 | 88.1 | 89.4 | 89.2 | 93.9 | 95.0 | 95.9 |
| CANE | 86.8 | 91.5 | 92.2 | 93.9 | 94.6 | 94.9 | 95.6 | 96.6 | 97.7 |
| DMTE (w/o diffusion) | 87.4 | 91.2 | 92.0 | 93.2 | 93.9 | 94.6 | 95.5 | 95.9 | 96.7 |
| DMTE (text only) | 82.6 | 84.0 | 85.7 | 87.3 | 89.1 | 91.1 | 92.0 | 92.9 | 94.2 |
| DMTE (Bi-LSTM) | 86.3 | 88.2 | 90.7 | 92.7 | 94.1 | 94.8 | 96.0 | 97.3 | 98.1 |
| DMTE (WAvg) | **91.3** | **93.1** | **93.7** | **95.0** | **96.0** | **97.1** | **97.4** | **98.2** | **98.8** |

Table 2: AUC scores for link prediction on Zhihu.

| % of edges | 15% | 25% | 35% | 45% | 55% | 65% | 75% | 85% | 95% |
|---|---|---|---|---|---|---|---|---|---|
| Deep Walk | 56.6 | 58.1 | 60.1 | 60.0 | 61.8 | 61.9 | 63.3 | 63.7 | 67.8 |
| LINE | 52.3 | 55.9 | 59.9 | 60.9 | 64.3 | 66.0 | 67.7 | 69.3 | 71.1 |
| node2vec | 54.2 | 57.1 | 57.3 | 58.3 | 58.7 | 62.5 | 66.2 | 67.6 | 68.5 |
| TADW | 52.3 | 54.2 | 55.6 | 57.3 | 60.8 | 62.4 | 65.2 | 63.8 | 69.0 |
| TriDNR | 53.8 | 55.7 | 57.9 | 59.5 | 63.0 | 64.6 | 66.0 | 67.5 | 70.3 |
| CENE | 56.2 | 57.4 | 60.3 | 63.0 | 66.3 | 66.0 | 70.2 | 69.8 | 73.8 |
| CANE | 56.8 | 59.3 | 62.9 | 64.5 | 68.9 | 70.4 | 71.4 | 73.6 | 75.4 |
| DMTE (w/o diffusion) | 56.2 | 58.4 | 61.3 | 64.0 | 68.5 | 69.7 | 71.5 | 73.3 | 75.1 |
| DMTE (text only) | 55.9 | 57.2 | 58.8 | 61.6 | 65.3 | 67.6 | 69.5 | 71.0 | 74.1 |
| DMTE (Bi-LSTM) | 56.3 | 60.3 | 64.9 | 69.8 | 73.2 | 76.4 | **78.7** | **80.3** | **82.2** |
| DMTE (WAvg) | **58.4** | **63.2** | **67.5** | **71.6** | **74.0** | **76.7** | 78.5 | 79.8 | 81.5 |

We set the embedding of dimension $d$ to 200 with $d_s$ and $d_t$ both equal to 100. The number of hops $H$ is set to 4 and the importance coefficients $\lambda_h$'s are tuned for different datasets and different tasks with $\lambda_0 > \lambda_1 > \cdots > \lambda_H$. $\alpha_{tt}$, $\alpha_{ss}$, $\alpha_{ts}$, and $\alpha_{st}$ are set to 1, 1, 0.3 and 0.3 respectively. The number of negative samples $K$ is set to 1 to speed up the training process. The word embedding matrix $\mathbf{E}_w$, the structure embedding table $\mathbf{E}_s$, and the diffusion weight matrix $\mathbf{W}$ are all randomly initialized with a truncated Gaussian distribution. All models are implemented in Tensorflow using a NVIDIA Titan X GPU with 12 GB memory.

## 5.1 Link Prediction

Given a pair of vertices, link prediction seeks to predict the existence of an unobserved edge using the trained representations. We use *Cora* and *Zhihu* datasets for link prediction. We randomly hold out a portion of edges ($\%e$) for training in an unsupervised way with the rest of edges for testing.

Tables 1 and 2 show the AUC scores of different models for $\%e$ from 15% to 95% on *Cora* and *Zhihu*. The best performance is highlighted in bold. As can be seen from both tables, our proposed method performs better than all other baseline methods. The AUC gains of DMTE model over the state-of-the-art CANE model can be as much as 4.5 and 6.8 on *Cora* and *Zhihu* respectively. These results demonstrate the effectiveness of the learned embeddings using the proposed method on link prediction task. We observe that baselines incorporating both structure and text

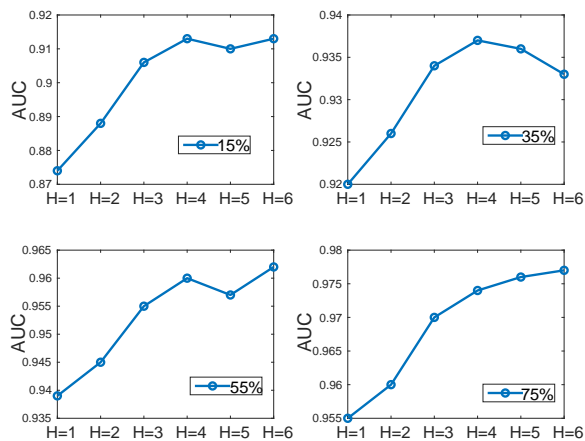

Figure 4: Performance over $H$.

Table 3: Top-5 similar vertex search based on embeddings learned by DMTE.

Query: The K-D-B-Tree: A Search Structure For Large Multidimensional Dynamic Indexes.
1. The R+-Tree: A Dynamic Index for Multi-Dimensional Objects.
2. The SR-tree: An Index Structure for High-Dimensional Nearest Neighbor Queries.
3. Segment Indexes: Dynamic Indexing Techniques for Multi-Dimensional Interval Data.
4. Generalized Search Trees for Database Systems.
5. High Performance Clustering Based on the Similarity Join.

information perform better than those only utilizes structure information, which indicates that text associated with each vertex helps to achieve more informative embeddings. The proposed approach shows flexibility and robustness in various training ratios. As the portion of training edges gets larger, the performance of our DMTE model steadily increases while other approaches suffer under either low training ratio (such as CENE) or high training ratio (such as TADW).

Comparing the four versions of DMTE, DMTE with word embedding average as the text inputs has the best performance on *Cora* at all training ratios and on *Zhihu* at low training ratios, while DMTE with bidirectional LSTM as the text inputs has the best performance on *Zhihu* at high training ratios. This is because when the training data is limited, the model with less parameters can successfully avoid over-fitting and thus achieve better results. For larger networks like *Zhihu* with high training data ratios, deep models (such as Bi-LSTM) with more parameters can be a good choice to encode input texts. The model with the diffusion convolutional operation applied on text inputs performs better than the model without the diffusion process, verifying our assumption that the diffusion process can help include long-distance semantic relationship and thus achieves better embeddings. We also observe that DMTE with text embeddings only performs better than some baseline methods but worse than the other three DMTE variations, demonstrating the effectiveness of text embeddings and the necessity of adding structure embeddings. Furthermore, DMTE with only the word-embedding average as the text representation has comparable performance over baselines, demonstrating the effectiveness of the redesigned objective function, which calculates the conditional probability of generating $v_i$ given the diffusion map of $v_j$.

**Parameter Sensitivity** Figure 4 shows the link prediction results w.r.t. the number of hops $H$ at different training ratios. The model we use here is DMTE(WAvg). Note that when $H = 1$ the model is equivalent to DMTE without diffusion precess. As $H$ gets larger, the performance of DMTE increases initially then stops increasing when $H$ is big enough. This observation indicates that the diffusion process can help exploit the relatedness of any two vertices in the graph, however this relatedness is neglectable when the distance between two vertices is too long.

## 5.2 Multi-Label Classification

Multi-label classification seeks to classify each vertex into a set of labels using the learned vertex representation as features. We use *DBLP* dataset for multi-label classification. Here DMTE refers to DMTE(WAvg). To maximally reduce the impact of complicated learning approaches on the classification performance, a linear SVM is employed instead of a sophisticated deep classifier. We randomly sample a portion of labeled vertices with embeddings ($\%l = \{10\%, 30\%, 50\%, 70\%\}$) to train the classifier with the rest vertices for testing.

Figure 5 shows the AUC scores of different models on *DBLP*. Compared to baselines, the proposed DMTE model consistently achieves performance improvement at all training ratios, demonstrating that DMTE learns high-quality embeddings which can be used directly as features for multi-label vertex classification. The F1-Macro score gains of DMTE over baseline

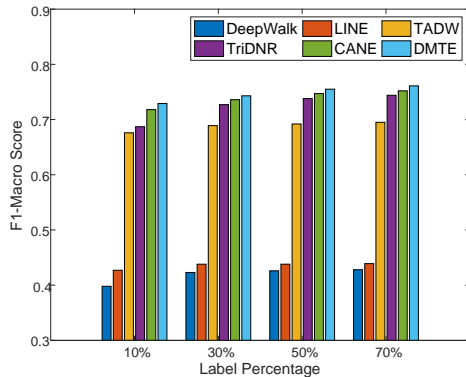

Figure 5: F1-Macro scores for multi-label classification on *DBLP*.

CANE indicates that the embeddings learned using global structure information is more informative than only considering local pairwise proximity. We also observe that structure-based methods perform much worse than methods based on structure and text combined, which further shows the importance of integrating both structure and text information in textual network embeddings.

## 5.3 Case Study

To visualize the effectiveness of the learned embeddings, we retrieve the most similar vertices and their corresponding texts for a given query vertex. The distance is evaluated by cosine similarity based on the vectorial representations learned by DMTE. Table 3 shows the texts of the top 5 closest vertex embeddings of a query paper in *DBLP* dataset. In the graph, vertices 1, 2, 4, and 5 are all neighbors of the query while vertex 3 is not directly connected with the query vertex. As observed, direct neighbors vertices 1 and 2 are not only structurally but also textually similar to the query vertex with multiple words aligned such as *tree*, *index* and *multi-dimensional*. Although vertex 3 is not on the same edge with the query vertex, the semantic relatedness makes it closer than the query's direct neighbors such as vertex 4 and 5. This is an illustration that the embeddings learned by DMTE successfully incorporate both structure and text information, helping to explain the quality of the aforementioned results.

# 6 Conclusions

We have proposed a new DMTE model for textual network embedding. Unlike existing embedding methods, that neglect semantic relatedness between texts or only exploit local pairwise relationship, the proposed method integrates global structural information of the graph to capture the level of connectivity between any two texts, by applying a diffusion convolutional operation on the text inputs. Furthermore, we designed a new objective that preserves high-order proximity, by including a diffusion map in the conditional probability. We conducted experiments on three real-word networks for multi-label classification and link prediction, and the associated results demonstrate the superiority of the proposed DMTE model.

**Acknowledgments**

The authors would like to thank the anonymous reviewers for their insightful comments. This research was supported in part by DARPA, DOE, NIH, ONR and NSF.

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
