[Reviews · NeurIPS 2018]

Reviewer 1



This paper considers the problem of textual network embedding. The authors adopt a diffusion map type method to incorporate both the graph structure information and the textual semantic information. The paper is clearly written and sophisticated with appropriate experiments following the demonstration of the methods. In both the multi-label classification task and the link prediction task, the proposed methods (with different variations) outperform the baselines in general. The choice of the datasets (Zhihu, Cora, and DBL) is also valid and representative. The parameter sensitivity analysis in the experiment section is also helpful and sometimes missing from similar works. I found the tuning of weights for the four objectives at (9)-(12) interesting. For others working with combined losses, usually the weighs are not essential. But in this work, I think the weights have some concrete meanings since each objective represents a particular part of the similarity between the original and embedded spaces. What does the fine-tuned weights look like? Any intuition? I have some technical questions for this work. Is there any intuition why the direct average of word embedding is better than Bi-LSTM embedding for most cases? Is there any particular reason why the authors include the concept of tensor in section 4.2? I am not able to find the necessity of doing so but it complicates the expression. === After rebuttal === Thanks for the rebuttal. I think the authors have answered most of my questions.

Reviewer 2



This paper presents a way to model a network of textual documents. To this end, the paper models the problem in vector space, where each edge (corresponding to a text document), the graph structure and the joint text and graph structure are all embedded in vector space. Text embeddings are modeled using word embeddings, and optionally using an RNN. The model parameters of the full network structure is learned together using a loss function that tries to generate neighboring vertices given a vertex. The authors measure performance on multi-label classification and link prediction tasks on various datasets. They get extremely good results across the board. I really like this paper and it presents a nice vector space take on the text-network modeling idea.

Reviewer 3



The main idea of this paper is to use the diffusion convolutional operator to learn text embedding that takes into account the global influence of the whole graph. It then incorporates the diffusion process in the loss function to capture high-order proximity. In contrast, previous works either neglect the semantic distance indicated from the graph, or fails to take into account the similarities of context influenced by global structural information. The author then conducts experiments on the task of multi-label classification of text and link prediction and shows that the proposed model outperforms the baselines. Strength: The high level idea of of this paper is good, and the method is novel. The diffusion convolutional operator is a good idea to softly incorporate the global factor of the whole graph. Weakness: The major quality problem of this paper is clarity. In terms of clarity, there are several confusing places in the paper, especially in equation 9, 10, 11, 12. 1) What is s_{i,j} in these equations? In definition 1, the author mentions that s_{i,j} denotes edge weights in the graph, but what are their values exactly in the experiments? Are they 0/1 or continuous values? 2) How is the diffusion map computed for structural embedding in 10 and 12? Is it using equation 1 only with the learned structural embedding and without text embedding? 3) Why is the diffusion convolution operator only applied to text embedding? Can it also be applied to structural embedding? On the other hand, if the author wants to capture global information in the graph as claimed between line 191 and line 194, why not directly use the diffusion map in equation (1) on text embedding instead of applying the diffusion convolution operator in 4.2? It's confusing to me the relationship between equation (1) and equation (5), (6) and (7) in section 4.2. In other words, there are two methods that could capture the global information: equation (1), and equation (5)(6)(7). Equation (1) is applied to structural embedding in equation (10) and (12); equation (5)(6)(7) are applied to textual embeddings. The author should explain why they do so. 4) I wonder whether the structural embedding is really necessary in this case, since in the learning process, the structural embedding just involves a embedding table lookup. The author does not explain the point of using a structural embedding, especially in such a way. What if just use diffusion text embedding? I don't see any experimental results proving the effectiveness of structural embedding in Table 1. 5) What's the motivation of each part in equation (8)? For example, what's the motivation of maximizing the probability of textual embedding of vertex i conditioned on the diffusion map of structural embedding of vertex j in equation (12)? 6) In line 135, the author says "Initially the network only has a few active vertices, due to sparsity." How is "active vertices" defined here? 7) In 5.2, when the author trains the SVM classifier, do they also fine-tune the embeddings or just freeze them? There are many existing CNN and RNN based neural classifier for text classification. What if just use any of those off-the-shelf methods on the text embedding without the diffusion process and fine-tune those embedding? This is typically a strong baseline for text classification, but there is no corresponding experimental results in Figure 5. 8) In line 234, how those objective function weights obtained? Are they tuned on any development set? Has the author tried only using a subset of those objectives? It's not clear how important each of those four objectives is. 9) In line 315, the author attributes the result of Table 3 to "both structure and text information". However, the fact that the method picks vertex 3 is due to diffusion convolution operator, as explained in line 313. Does the "structure" here mean the diffusion convolution operator or the structural embedding?